# Comparison of Three Dietary Assessment Methods to Estimate Meat Intake as Part of a Meat Reduction Intervention among Adults in the UK

**DOI:** 10.3390/nu14030411

**Published:** 2022-01-18

**Authors:** Cristina Stewart, Filippo Bianchi, Kerstin Frie, Susan A. Jebb

**Affiliations:** Nuffield Department of Primary Care Health Sciences, University of Oxford, Oxford OX2 6GG, UK; filippo.bianchi@bi.team (F.B.); kfrie.research@gmail.com (K.F.); susan.jebb@phc.ox.ac.uk (S.A.J.)

**Keywords:** meat consumption, dietary assessment, food diary, dietary recall

## Abstract

Food diaries are used to estimate meat intake at an individual level but it is unclear whether simpler methods would provide similar results. This study assessed the agreement between 7 day food diaries in which composite dishes were disaggregated to assess meat content (reference method), and two simpler methods: (1) frequency meal counts from 7 day food diaries; and (2) 7 day dietary recalls, each using standard estimated portion sizes. We compared data from a randomized controlled trial testing a meat reduction intervention. We used Bland-Altman plots to assess the level of agreement between methods at baseline and linear mixed-effects models to compare estimates of intervention effectiveness. At baseline, participants consumed 132 g/d (±75) of total meat; frequency meal counts and dietary recalls underestimated this by an average of 30 and 34 g/day, respectively. This was partially explained by an underestimation of the assumed portion size. The two simpler methods also underestimated the effect of the intervention, relative to control, though the significant effect of the intervention was unchanged. Simpler methods underestimated absolute meat intake but may be suitable for use in studies to measure the change in meat intake in individuals over time.

## 1. Introduction

Meat intake, particularly red and processed meat, negatively affects human health, and meat production negatively impacts the environment [1,2]. Accurately and reliably measuring meat consumption within the population is important to quantify current levels of consumption, identify trends in consumption over time, and assess the extent to which interventions aiming to reduce meat consumption effectively promote this dietary change. However, assessing dietary intake is difficult. Most diet assessments of free-living participants rely on self-reported measures (e.g., food diaries, dietary recalls and food frequency questionnaires (FFQs)), with each method having its own strengths, limitations and level of investigator and participant burden [3,4].

Food diaries, both weighed and un-weighed (estimated), have been the mainstay of dietary assessment for many years, including in the UK National Diet and Nutrition Survey (NDNS) [5], the EPIC Norfolk Study [6], the Avon Longitudinal Study of Parents and Children cohort [7] and the MRC National Survey of Health and Development (1946 British Birth Cohort) [8]. Respondents record all of the food and beverages they consume, prospectively, over a specific period [9]. Diaries can provide detailed intake data if completed thoroughly as a contemporaneous record. However, they rely on the respondent being motivated, compliant, numerate and literate and incur a large respondent burden. Food diaries are also time-consuming and costly for researchers; respondents need to be trained on how to use their diaries effectively and composite meat products or dishes need to be disaggregated into component ingredients [10]. In 2008, the NDNS moved to a 4 day estimated diary to reduce participant burden after previously using 7 day weighed diaries [5].

Recall-based methods, such as 24 h dietary recalls and FFQs, which ask respondents to report information retrospectively about food consumed over a specific period of time, are also commonly used [11,12]. They can be interviewer administered or web based, and in comparison to disaggregating 7 day food diaries, they carry a considerably lower burden for both investigators and participants. However, these methods may be more prone to recall bias and intentional misreporting [4]. To help researchers select the most appropriate method to collect dietary data on meat consumption, it is important to understand whether simpler dietary assessment methods, which impose a lower burden on respondents and investigators, would yield similar results to that of food diaries.

The current study aimed to assess the agreement between methods to estimate meat consumption between disaggregated 7 day food diaries and two simpler frequency methods in a population of UK adult volunteers taking part in a randomized controlled trial (RCT) of an intervention to reduce meat consumption [13,14]. The two simpler approaches assessed were frequency meal counts from 7 day food diaries and 7 day dietary recalls, each using standard estimated portion sizes to estimate meat intake in g/day from these frequency measures (methods are described below in more detail). As consumption of different meat types has different effects on health and the natural environment, we also aimed to assess the agreement between methods to estimate intake of (i) red and processed meat, and (ii) white meat, separately.

## 2. Materials and Methods

### 2.1. Participants

Dietary data were obtained from 115 participants who took part in RE-MAP (Replacing Meat with Alternative Plant-based Products), an RCT evaluating the impact of a four-week behavioural intervention to reduce meat consumption [14]. RE-MAP was conducted in Oxford, United Kingdom, and participants were recruited through community advertisements. Participants were eligible if they belonged to adult-only households, self-reported to eat meat at least five times per week and did not eat meat alternatives regularly. Eligible participants were invited to attend an enrolment appointment where written informed consent was collected, and participants were trained on how to keep an accurate food diary and best estimate portion sizes. Participants were then randomized 1:1 to intervention or control groups. RE-MAP was granted ethical approval by the Medical Sciences Interdivisional Research Ethics Committee (IDREC) of the University of Oxford (REF: R54329/RE001).

### 2.2. Dietary Measurements

Following enrolment, participants completed a prospective 7 day estimated un-weighed food diary leading up to their baseline appointment. At this appointment, participants completed a dietary recall questionnaire that asked them to, retrospectively, recall how many times they had eaten meat in the preceding 7 days. Both dietary measurements were repeated at the 4 week (at the end of the intervention) and 8 week (four weeks after intervention completion) follow-ups. The study process and methodology have been described in full previously [13,14].

At each time point, we estimated participants’ mean total meat intake in g/day using three methods:
Disaggregating the quantity of meat from meat-containing composite products recorded in 7 day food diaries (high investigator/participant burden);Counting the frequency of meals containing meat recorded in 7 day food diaries, multiplied by a standard portion size of meat (medium investigator burden, high participant burden);Asking participants to retrospectively recall how many times they consumed meat in the preceding 7 days through a questionnaire, multiplied by a standard portion size of meat (low investigator/participant burden).

The standard portion sizes of meat used to transform our frequency measures into g/day were obtained from a specific meat frequency questionnaire, which utilized portion size information from the UK Food Standards Agency and meat disaggregation data from the food composition database of the UK’s NDNS [15].

We estimated participants’ mean daily intake of red and processed meat, and unprocessed white meat, separately, using the same approach. We used the World Health Organization’s International Agency for Research on Cancer’s definition for red and processed meat [16]. That is, red meat comprised all unprocessed beef, veal, pork, lamb, mutton, and goat, and processed meat included meat that had been transformed through salting, curing, fermentation, smoking or other processes to enhance flavour or improve preservation (e.g., sausages, bacon, and ham). For the purpose of this study, breaded and battered meat products (e.g., chicken nuggets) were also grouped with processed meat. Unprocessed white meat included poultry (e.g., chicken, turkey, goose, and duck) without processing (with the exception of basic mincing). Unprocessed game meat (e.g., guinea fowl, pheasant, and rabbit) was also grouped with ‘unprocessed white meat’ as game meat consumption was negligible within this study population.

### 2.3. Disaggregated Food Diaries

Participants used MyFitnessPal, an electronic smartphone application, to record their food intake over 7 days, estimating the weight in grams or household measures (e.g., units, cups). This app allowed users to add foods to their diaries either manually or by scanning barcodes. Participants received daily text messages reminding them to complete their diaries and were asked to complete them prospectively.

Meat consumption was estimated in mean g/day by disaggregating meat-containing composite products in the food diaries. The disaggregation process involved four steps: (i) estimating the weight of the whole product; (ii) estimating the proportion of meat; (iii) converting the weight of uncooked meat to cooked meat, where applicable; and (iv) categorizing meat into the different sub-types of meat. We categorized meat into red and processed meat, and unprocessed white meat and calculated a total sum. We assumed this method to be the most accurate and therefore employed it as the reference method within this study.

### 2.4. Frequency Meal Counts from Food Diaries

Participants could enter a maximum of six possible meal occasions per day (breakfast, mid-morning, lunch, mid-afternoon, dinner, and post-dinner) into their food diaries. The consumption frequency of meat was measured by counting the number of meal entries containing meat (ranging from 0 to 6 per day) and calculating daily mean values for baseline, 4 week and 8 week follow-ups. We transformed this frequency measure into g/day by multiplying the mean number of meals containing meat by standard portion sizes (total meat 69.3 g; red and processed meat 64.1 g; unprocessed white meat 80.3 g), obtained from a specific meat frequency questionnaire [15].

### 2.5. Retrospective 7 Day Dietary Recalls

Mean daily meat consumption frequency was also estimated through 7 day dietary recall questionnaires (at baseline, 4 week, and 8 week follow-ups). Participants were asked to recall the number of times they consumed meat, over the same week as the food diary. This frequency measure was transformed into g/day using the same methodology as the one we employed for frequency meal counts.

### 2.6. Statistical Analysis

#### 2.6.1. Baseline Comparisons

Agreement between each paired measure at baseline was assessed with Bland-Altman plots, based on the mean difference and 95% limits of agreement (LOA) between the dietary assessment methods [17]. Linear regressions were employed to assess the relationship between the bias and magnitude of the measurement [18].

We calculated average portion sizes for total meat, red and processed meat, and unprocessed white meat from the food diaries by dividing the mean daily disaggregated consumption by the mean daily frequency meal count, for each respective category. We then descriptively compared these with the standard portion sizes utilized for the two frequency dietary measures.

#### 2.6.2. Difference in Intervention Effectiveness Analyses

To determine whether the difference in the change in meat consumption at the 4 and 8 week follow-ups measured between the intervention and the control group differed depending on the dietary assessment method employed, we used linear mixed-effects models.

Fixed effects were included for randomized group, follow-up visit, the interaction between follow-up visit and randomized group, baseline meat consumption and sex. Random effects for participants’ intercept and slope were included to account for repeated measures on the same participant. The adjusted treatment effects were reported with their 95% confidence intervals. We undertook the same analysis using the three different measures to descriptively explore whether the impact of the intervention on meat consumption differed depending on the method employed to measure consumption of meat. Statistical analyses were performed using Stata/IC version 14.1(StataCorp LLC, College Station, TX, USA). A *p* value < 0.05 was set to denote statistical significance.

## 3. Results

### 3.1. Demographics

The mean age of participants was 35 (±11) years, 65% were female and 57% were British. Forty-four percent of participants had obtained a Bachelor or equivalent as their highest degree and 41% had obtained a higher degree (Table 1).

### 3.2. Baseline Comparisons

At baseline, participants consumed 132 g/d (±75) of total meat when measured by disaggregated food diaries (reference). Frequency meal counts and dietary recalls underestimated meat intake by an average of 30 and 34 g/day, respectively (Table 2). Comparisons of the average portion sizes calculated from our food diaries, with the standard portion sizes underlying the two frequency measures, showed that the latter underestimated total meat by 20.2 g per serving (Appendix A).

Although both frequency meal counts and dietary recalls underestimated total meat intake in comparison to our reference method, they showed good agreement with each other (Figure 1). The mean absolute difference (bias) in total meat intake estimated between frequency meal counts and dietary recalls was 4.0 g/day (95% LOA −86 to 94 g/day; Figure 1). Regression analysis showed significantly greater underestimation of meat intake at higher intakes using both frequency methods compared to our reference method. At higher levels of consumption, the frequency meal count method underestimated relative to dietary recalls (Table 3; Figure 1). Bland-Altman plots for red and processed meat and unprocessed white meat were similar (Appendix A). Looking at specific meat types, we found that the average portion sizes attributed to each eating occasion for red and processed meat was 19.7 g lower for dietary recalls and frequency meal counts than food diaries, while that of unprocessed white meat was 21.2 g lower (Appendix A).

### 3.3. Comparison of Intervention Effects

Total meat consumption at the 4 and 8 week follow-ups by study group is presented in Table 4. Frequency meal counts underestimated the reduction in consumption by 17 and 8 g/day, and dietary recalls by 20 and 16 g/day at the 4 week and 8 week follow-ups, respectively. However, the 95% confidence intervals were overlapping and the interpretation of the significant effect of the intervention on meat intake was unchanged (Table 4; Figure 2).

Results from mixed-effects model adjusting for baseline meat consumption and sex. Data are the mean (g/day) and 95% confidence intervals. *N* = 114 at four weeks and *N* = 113 at eight weeks. Disaggregated food diaries: disaggregating the quantity of meat from meat-containing composite products recorded in 7 day food diaries; frequency meal counts: counting the frequency of meals containing meat recorded in 7 day food diaries*standard portion size; dietary recalls: asking participants to retrospectively recall how many times they consumed meat in the last 7 days through a questionnaire*standard portion size.

## 4. Discussion

This study assessed the agreement between two frequency dietary assessment methods for estimating meat intake among UK adults taking part in an RCT of an intervention to reduce meat consumption—a frequency count of meat servings based on a 7 day food diary, and a 7 day recall of meat intake—and our reference method, 7 day food diaries with composite dishes fully disaggregated to identify meat content. We showed that the two simpler methods had very good agreement with each other but underestimated total meat intake compared to disaggregated food diaries. Moreover, they both underestimated the effect of the intervention, relative to control, though the significant effect of the intervention was unchanged. This can likely be attributed to the underlying standard portion sizes of meat being too small to accurately represent consumption in this population.

In comparison to our reference method, frequency meal counts performed slightly better than dietary recalls in measuring total meat intake and change in intake over time—which might be because frequency meal counts were based on the same food diaries that were used to create disaggregated food diaries. However, at high levels of intake, this relationship was inverted, with frequency meal counts underestimating total meat intake more than dietary recalls. Although dietary recalls are usually considered more prone to recall bias [9], previous studies estimating the differences in macro-and micro-nutrient intakes have found dietary recalls to be less affected by systemic errors than FFQs [19]. Moreover, another explanation for the dietary recalls performing better at higher levels of meat intake is that our dietary recalls were meat-specific while the frequency meal counts were made from food diaries covering the whole diet. We speculate that a meat-specific food diary may perform better and that asking participants to indicate each occasion they consume meat, as opposed to recording every food and beverage consumed, would also reduce participant burden [20]. Indeed, a previous study has reported that respondents preferred a food checklist comprised of a pre-printed list of foods that they could tick when eaten, over dietary recalls and food diaries, because they felt they were easier to complete [21].

Dietary reporting error differs across nutrients, and our study is focused specifically on meat [19]. To the best of our knowledge, no study to date has compared frequency meal counts or dietary recalls to estimate intakes of specific food groups [22]. One previous study evaluated the consistency of dietary patterns between two dietary recalls (24 and 48 h) and a 5 day estimated food diary in a British population [23]. Concurring with our findings, they found no significant differences in intakes of individual meat groups.

Strengths of this study include the collection of dietary data from 3 weeks, each covering 7 consecutive days, which controls for day-to-day variation. High satisfaction with, and preference for, smartphone technology to undertake dietary assessments have been reported previously and the use of a commercially available smartphone application to collect food diary data may be less obtrusive and burdensome for respondents than paper diaries [24]. Moreover, studies comparing MyFitnessPal with paper 7 day food diaries have found it provides accurate estimates of energy intake though may underestimate intake of some macro- and micro-nutrients [25,26]. This is the first study to explore the comparability of three different meat assessment measures when employed both as dietary surveys and to measure the impact of a meat reduction intervention. Moreover, this is the first study to utilize frequency meal counts and dietary recalls as a tool to estimate meat consumption.

There are also limitations to the current study that should be acknowledged. Firstly, the RE-MAP study participants may not have been representative of the adult UK population. Over three-quarters of participants (*n* = 98, 85%) held a university or higher degree, compared with 27% of those ≥16 years in the UK-wide population [27]. This is likely attributed to the fact that recruitment was carried out through public advertisements in Oxford. Secondly, participants completed the dietary recall after keeping a 7 day food diary, which may have led to a substantial learning effect. Accordingly, the results of our dietary recall may not be indicative of how the method will perform in future research when this method is employed alone. Nevertheless, the two simpler frequency methods we employed here had good agreement with each other and we can speculate that a meat-specific checklist may be suitable to measure change in meat intake in individuals over time. Moreover, utilizing more accurate standard portion sizes of meat to transform frequency measures into g/day may improve estimates of meat intake in both frequency measures. We consider that this study is valuable as a rare opportunity to look at the agreement between methods over the course of an intervention study; however, such studies will usually have a smaller sample size than observational analyses. Given we only assessed the level of agreement between methods in this study, future research may wish to test the reliability and validity of the two simpler methods tested here.

## 5. Conclusions

Our results showed the disaggregated food diaries provide higher absolute estimates of meat intake in the context of a meat reduction intervention than frequency meal counts and dietary recalls. The two simpler methods underestimated the effect of the intervention, relative to control, though the significant effect of the intervention was unchanged. These simpler methods may be a practical option to rank participants in dietary surveys and to measure changes over time. However, the estimates of meat intake for these measures could be improved with more accurate portion size estimates, perhaps combined with a meat-specific recall questionnaire.

## Figures and Tables

**Figure 1 nutrients-14-00411-f001:**
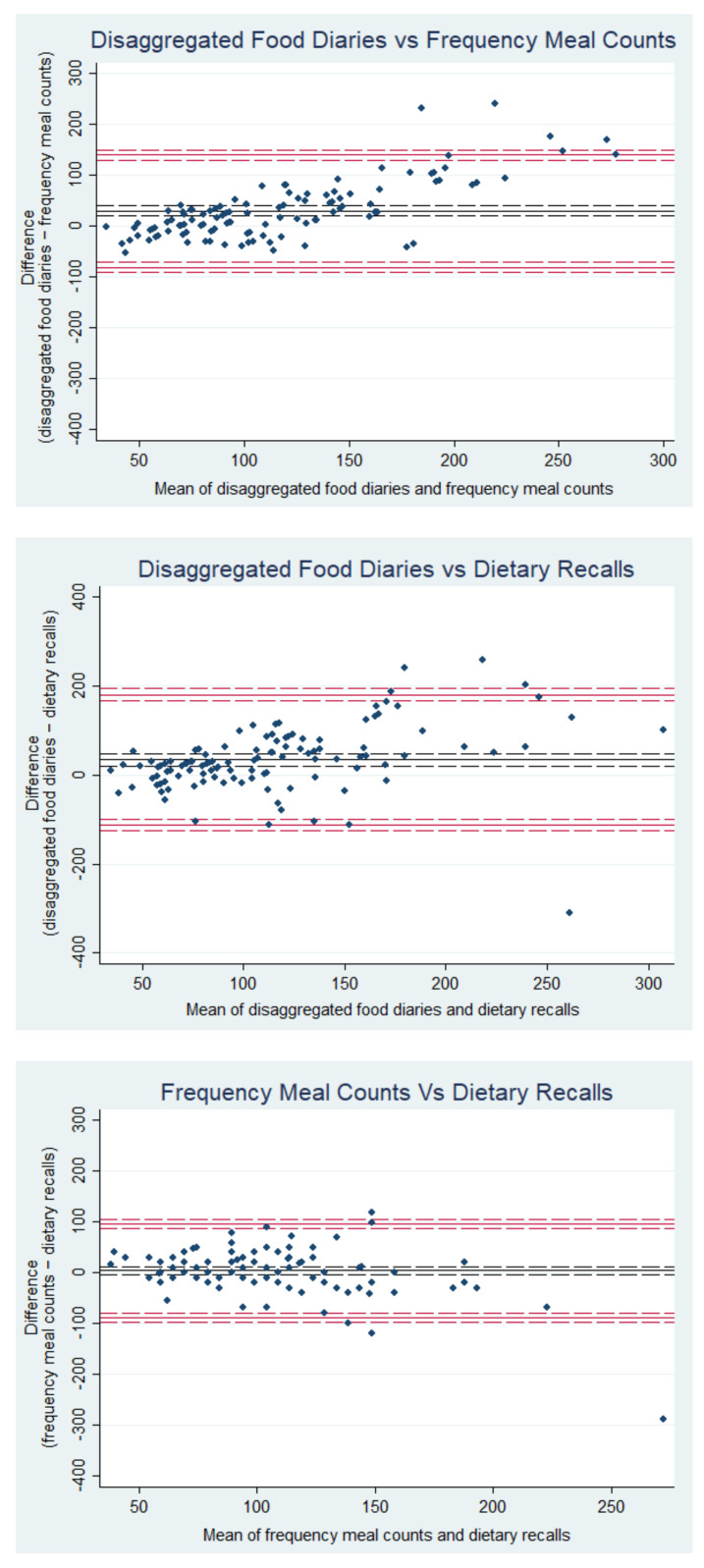
Bland-Altman plots showing the comparability of three methods to estimate meat intake (g/day). Disaggregated food diaries: disaggregating the quantity of meat from meat-containing composite products recorded in 7 day food diaries; frequency meal counts: counting the frequency of meals containing meat recorded in 7 day food diaries*standard portion size; dietary recalls: asking participants to retrospectively recall how many times they consumed meat in the last 7 days through a questionnaire*standard portion size. Solid red lines are the limits of agreement with 95% confidence intervals (red dashed lines). The solid black line is the mean difference (bias) together with the 95% confidence intervals (black dashed lines).

**Figure 2 nutrients-14-00411-f002:**
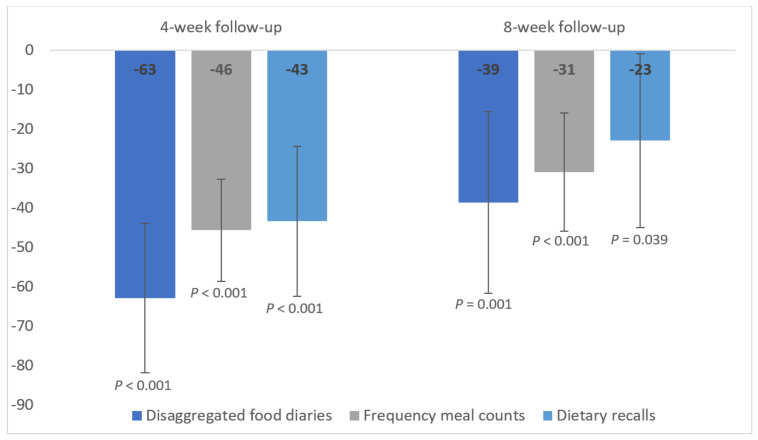
Difference in g/day of total meat consumption relative to control. Mixed-effects models with fixed effects for randomized group, baseline meat consumption and sex, and random effects for participants’ intercept and slope. Values are the mean and 95% confidence intervals. *N* = 114 at four weeks and *N* = 113 at eight weeks. Disaggregated food diaries: disaggregating the quantity of meat from meat-containing composite products recorded in 7 day food diaries; frequency meal counts: counting the frequency of meals containing meat recorded in 7 day food diaries*standard portion size; dietary recalls: asking participants to retrospectively recall how many times they consumed meat in the last 7 days through a questionnaire*standard portion size.

**Table 1 nutrients-14-00411-t001:** Baseline demographic characteristics.

	Control (*N* = 57)	Intervention (*N* = 58)
Age in Years	37 (12)	33 (10)
Gender		
Female	37 (65%)	38 (66%)
Male	19 (33%)	19 (33%)
Other/prefer not to say	1 (2%)	1 (2%)
Ethnic origin		
White	45 (79%)	50 (86%)
Chinese	2 (4%)	4 (7%)
Black African/Caribbean	1 (2%)	1 (2%)
Other/prefer not to say	9 (16%)	3 (5%)
Highest educational degree		
GCSE or equivalent	1 (2%)	4 (7%)
A-level or equivalent	5 (9%)	7 (12%)
BSc or equivalent	23 (40%)	28 (48%)
Higher degree (MSc, PhD or equivalent)	28 (49%)	19 (33%)

Data are the mean and standard deviation or number and percentage.

**Table 2 nutrients-14-00411-t002:** Baseline meat consumption (g/day).

	Disaggregated Food Diaries			Frequency Meal Counts			Dietary Recalls		
	Total	Control	Intervention	Total	Control	Intervention	Total	Control	Intervention
Total meat	132 (75)	134 (72)	130 (78)	102 (37)	102 (39)	101 (34)	98 (53)	102 (45)	94 (59)
Red and processed meat	83 (62)	82 (57)	84 (66)	64 (32)	63 (34)	64 (31)	61 (41)	65 (40)	57 (42)
Unprocessed white meat	49 (38)	51 (38)	46 (37)	38 (24)	39 (21)	37 (25)	37 (29)	36 (25)	37 (32)

Data are the mean meat consumption g/day and standard deviation. Disaggregated food diaries: disaggregating the quantity of meat from meat-containing composite products recorded in 7 day food diaries; frequency meal counts: counting the frequency of meals containing meat recorded in 7 day food diaries*standard portion size; dietary recalls: asking participants to retrospectively recall how many times they consumed meat in the last 7 days through a questionnaire*standard portion size.

**Table 3 nutrients-14-00411-t003:** Linear regression analyses for magnitude bias for estimating meat intake obtained by disaggregated food diaries, frequency meal counts and dietary recalls.

	Mean Difference (bias) in g/day	Regression Coefficient	*p*-Value	95% CI
Disaggregated food diaries vs. frequency meal counts	29.8	0.78	<0.001	0.71, 0.86
Disaggregated food diaries vs. dietary recalls	33.8	0.50	<0.001	0.36, 0.63
Frequency meal counts vs. dietary recalls	4.0	−0.47	<0.001	−0.58, −0.35

Disaggregated food diaries: disaggregating the quantity of meat from meat-containing composite products recorded in 7 day food diaries; frequency meal counts: counting the frequency of meals containing meat recorded in 7 day food diaries*standard portion size; dietary recalls: asking participants to retrospectively recall how many times they consumed meat in the last 7 days through a questionnaire*standard portion size. CI; Confidence Interval.

**Table 4 nutrients-14-00411-t004:** Adjusted total meat consumption (g/day) at four- and eight-week follow-ups.

	Control		Intervention		Difference between Groups	
	Mean	95% CI	Mean	95% CI	Mean	95% CI
Disaggregated food diaries						
4 weeks	115	102, 128	52	39, 66	−63	−82, −44
8 weeks	120	104, 137	82	66, 98	−39	−62, −16
Frequency meal counts						
4 weeks	88	79, 97	43	34, 52	−46	−58, −33
8 weeks	95	84, 105	64	53, 74	−31	−46, −16
Dietary recalls						
4 weeks	86	72, 99	42	29, 56	−43	−63, −24
8 weeks	91	75, 107	67	51, 83	−23	−46, −1

## Data Availability

The data presented in this study are available from the corresponding author on reasonable request.

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
