# Peer review of "Comparison of Three Dietary Assessment Methods to Estimate Meat Intake as Part of a Meat Reduction Intervention among Adults in the UK"

_nutrients, 2022, doi:10.3390/nu14030411_

Round 1

Reviewer 1 Report

Congratulations on the study, which is of great interest.

As a comment of possible improvement, I think it would be important to provide some data or reference of possible validation studies of the MyFitnessPal App in interventions to quantify food consumption.

Best regards, the reviewer.

Author Response

Congratulations on the study, which is of great interest.

As a comment of possible improvement, I think it would be important to provide some data or reference of possible validation studies of the MyFitnessPal App in interventions to quantify food consumption.

Best regards, the reviewer.

Thank you very much for your feedback. We have now referenced two studies that validated MyFitnessPal against paper food diaries. We have added the following text to the discussion (lines 287-289): “Moreover, studies comparing MyFitnessPal with paper 7-day food diaries have found it provides accurate estimates of energy intake but may underestimate intake of some macro- and micro-nutrients”

Reviewer 2 Report

Interesting work, but it requires editing. Authors should pay particular attention to Materials and Methods and the Introduction section. Introduction

Lines: 47-48. Please explain the excerpt: …… other dietary assessment methods, which impose a lower burden on respondents and investigator ……

Line 50: Please explain and discuss the two simpler frequency methods. Later in work, the authors consider dividing meat into white, red and processed meat.

I propose to discuss what the concept of white, red and processed meat means and based on what criterion this division is.

Lines: 53-58. Add the number of study participants.

Lines: 62-64. Please explain.

Line 70. What does: “standard portion size of meat”.

Lines: 76-77. On what basis "unprocessed game meat was grouped with" unprocessed white meat ". Therefore, I propose to include the division of meat into white and red in the Introduction section.

Line 99. Fragment unclear (baseline, 4-week and 8-week follow-ups).

Lines: 109-111. Please explain. Table 1. Did materials and methods find a division into control group and intervention? Were an interview conducted when entering the study? Do the respondents limit meat in their diet? Does meat consumption differ by gender? The same is true for ethnic origin. Why did the authors perform a demographic assessment of the respondents? Is not their number too low to conclude the methods to estimate meat intake?

Lines: 238. In the reviewer's opinion, the limitations indicated in this fragment make it impossible to obtain reliable data. It is therefore advisable to deepen the study. 

Author Response

Responses to reviewer 2

Interesting work, but it requires editing. Authors should pay particular attention to Materials and Methods and the Introduction section.

We have reviewed the whole text and made small edits which we hope make our message clearer.

Point 1. Lines: 47-48. Please explain the excerpt: …… other dietary assessment methods, which impose a lower burden on respondents and investigator ……

We apologise that this was unclear. We have edited the sentence to emphasise that food diaries were our reference method (lines 57-58): “[…] simpler dietary assessment methods, which impose a lower burden on respondents and investigators, would yield similar results to that of food diaries.”

Point 2. Line 50: Please explain and discuss the two simpler frequency methods. Later in work, the authors consider dividing meat into white, red and processed meat. I propose to discuss what the concept of white, red and processed meat means and based on what criterion this division is.

Thank you for this suggestion. We have now added the following to the introduction (lines 62-68): “The two simpler approaches assessed were frequency meal counts from 7-day food diaries and 7-day dietary recalls, each using standard estimated portion sizes to estimate meat intake in g/day from these frequency measures (methods are described below in more detail). As consumption of different meat types has different effects on health and the natural environment, we also aimed to assess the agreement between methods to estimate: (i) red & processed meat, and (ii) white meat, separately.”

In the materials and methods section, we have provided more detail on how we categorised the individual meat types (lines 111-123): “We used the World Health Organization (WHO)’s International Agency for Research on Cancer’s definition for red and processed meat [16]. That is, red meat comprised all unprocessed beef, veal, pork, lamb, mutton, and goat, and processed meat included meat that had been transformed through salting, curing, fermentation, smoking or other processes to enhance flavor or improve preservation (e.g. sausages, bacon, and ham). For the purpose of this study, we also classed breaded and battered meat products (e.g. chicken nuggets) as processed meat. Unprocessed white meat included poultry (e.g. chicken, turkey, goose, and duck) without processing (with the exception of basic mincing). Unprocessed game meat (e.g. guinea fowl, pheasant, and rabbit) was also grouped with ‘unprocessed white meat’ as game meat consumption was negligible within this study population.”

Point 3. Lines: 53-58. Add the number of study participants.

We have now added this on line 72.

Point 4. Lines: 62-64. Please explain.

We have added further clarification to this sentence on lines 86-93 which now reads “Following enrolment, participants completed a prospective 7-day estimated un-weighed food diary leading up to their baseline appointment. At this appointment, participants completed a dietary recall questionnaire that asked them to, retrospectively, recall how many times they had eaten meat in the preceding 7 days. Both dietary measurements were repeated at the 4-week (at the end of the intervention) and 8-week (four weeks after intervention completion) follow-ups. The study process and methodology has been described in full previously.”

Point 5. Line 70. What does: “standard portion size of meat”.

Apologies this was not clear from the outset. We had explained this in section 2.4 of the manuscript (“frequency meal counts from food diaries”) but have now moved this earlier in the paper. We have added the following text to lines 106-109: “The standard portion sizes of meat used to transform our frequency measures into g/day were obtained from a specific meat frequency questionnaire, which utilized portion size information from the UK’s Food Standards Agency and meat disaggregation data from the food composition database of the UK’s NDNS”

Point 6. Lines: 76-77. On what basis "unprocessed game meat was grouped with" unprocessed white meat ". Therefore, I propose to include the division of meat into white and red in the Introduction section.

Thank you for this suggestion. As in our response to your second point, we have added the following text to the introduction on lines 65-68: “As consumption of different meat types has different effects on health and the natural environment, we also aimed to assess the agreement between methods to estimate: (i) red & processed meat, and (ii) white meat, separately.”

And the following in the materials and method section to lines 110-123:

“We estimated participants’ mean daily intake of red & processed meat, and unprocessed white meat, separately, using the same approach. We used the World Health Organization (WHO)’s International Agency for Research on Cancer’s definition for red and processed meat. That is, red meat comprised all unprocessed beef, veal, pork, lamb, mutton, and goat, and processed meat included meat that had been transformed through salting, curing, fermentation, smoking or other processes to enhance flavor or improve preservation (e.g. sausages, bacon, and ham). For the purpose of this study, we also classed breaded and battered meat products (e.g. chicken nuggets) as processed meat. Unprocessed white meat included poultry (e.g. chicken, turkey, goose, and duck) without processing (with the exception of basic mincing). Unprocessed game meat (e.g. guinea fowl, pheasant, and rabbit) was also grouped with ‘unprocessed white meat’ as game meat consumption was negligible within this study population.”

In reality, the amount of unprocessed game meat consumed was minimal (mean intake of 1 g/day at baseline) and so this classification does not meaningfully affect the results.

Point 7. Line 99. Fragment unclear (baseline, 4-week and 8-week follow-ups).

We hope the text added in response to the previous point 4 has provided the necessary clarification. That is, on lines 90-93 we added the following text “Both dietary measurements were repeated at the 4-week (at the end of the intervention) and 8-week (four weeks after intervention completion) follow-ups. The study process and methodology has been described in full previously”

Point 8. Lines: 109-111. Please explain. Table 1. Did materials and methods find a division into control group and intervention? Were an interview conducted when entering the study? Do the respondents limit meat in their diet? Does meat consumption differ by gender? The same is true for ethnic origin. Why did the authors perform a demographic assessment of the respondents? Is not their number too low to conclude the methods to estimate meat intake?

Thank you very much for your comment. We have now added further details regarding the RE-MAP study participants on lines 75-82 which we hope sufficiently addresses your queries:

“RE-MAP was conducted in Oxford, United Kingdom, and participants were recruited through community advertisements. Participants were eligible if they belonged to adult-only households, self-reported to eat meat at least five times per week and did not eat meat alternatives regularly. Eligible participants were invited to attend an enrolment appointment where written informed consent was collected, and participants were trained on how to keep an accurate food diary and best estimate portion sizes. Participants were then randomized 1:1 to intervention or control groups.”

Table 1 describes baseline demographic characteristics of our participants to provide readers with this information. We did not set out here to explore whether meat consumption differed by gender or ethnicity as this would require a much larger survey. The focus of our study was the level of agreement between methods to estimate meat intake. However, we adjusted for sex in our linear mixed effects models assessing the change in meat intake from baseline to both follow-ups, as stated on lines 173-174.

Differences in meat consumption between the intervention and control group have been reported in the main trial paper: https://doi.org/10.1093/ajcn/nqab414 

Point 9. Lines: 238. In the reviewer's opinion, the limitations indicated in this fragment make it impossible to obtain reliable data. It is therefore advisable to deepen the study. 

Thank you for raising your concerns. We have tried to be clear about the limitations of this study, but as this is a secondary data analysis of an already published RCT, we are unable to expand the study. We have expanded our limitations section and added the following to lines 302-309: “Nevertheless, the two simpler frequency methods we employed here had good agreement with each other and we can speculate that a meat-specific checklist may be suitable to measure change in meat intake in individuals over time. Moreover, utilizing more accurate standard portion sizes of meat to transform frequency measures into g/day may improve estimates of meat intake in both frequency measures. We consider that this study is valuable as a rare opportunity to look at the agreement between methods over the course of an intervention study, however, such studies will usually have a smaller sample size than observational analyses.

Reviewer 3 Report

Article with the title "Comparison of three dietary assessment methods to estimate meat intake as part of a meat reduction intervention among adults in the UK" is very interesting.

In the introduction, the authors provide sufficient background. Article meets usual requirements also in the next chapters. However, I recommend including at least two references more in introduction within five years of their publication.

The authors clearly and comprehensibly specified the objectives of the experiment, explained the background and methods used, and summarized all the essentials in the conclusion. I find this article interesting for readers, despite its simplicity and relative brevity, it will just have the potential to be cited by interested authors. So, I only recommend expanding the current articles of other authors in the introduction section.

Author Response

In the introduction, the authors provide sufficient background. Article meets usual requirements also in the next chapters. However, I recommend including at least two references more in introduction within five years of their publication.

The authors clearly and comprehensibly specified the objectives of the experiment, explained the background and methods used, and summarized all the essentials in the conclusion. I find this article interesting for readers, despite its simplicity and relative brevity, it will just have the potential to be cited by interested authors. So, I only recommend expanding the current articles of other authors in the introduction section.

Thank you for taking the time to review our paper. We have now added the following additional information into our introduction and cited four additional references:

“Most diet assessments of free-living participants rely on self-reported measures (e.g. food diaries, dietary recalls and food frequency questionnaires (FFQs)), with each method having its own strengths, limitations and level of investigator and participant burden.” (Lines 32-35)

“Recall-based methods, such as 24-hour dietary recalls and FFQs, which ask respondents to report information retrospectively about food consumed over a specific period of time, are also commonly used. They can be interviewer-administered or web-based, and in comparison to disaggregating 7-day food diaries, they carry a considerably lower burden for both investigators and participants. However, these methods may be more prone to recall bias and intentional misreporting [4]. (Lines 49-54)

The references added were:

  1. NIHR/MRC. Diet, Anthropometry and Physical Activity (DAPA) Measurement Toolkit. Available online: https://dapa-toolkit.mrc.ac.uk/diet/subjective-methods/24-hour-dietary-recall (accessed on 6 January 2022).
  2. Naska A, Lagiou A, Lagiou P. Dietary assessment methods in epidemiological research: current state of the art and future prospects. F1000Res. 2017, 6, 926. doi: 10.12688/f1000research.10703.1
  3. Bradbury, K,E.; Young, H,J.; Guo, W.; Key, T,J. Dietary assessment in UK Biobank: an evaluation of the performance of the touchscreen dietary questionnaire. J Nutr Sci. 2018, 7, doi: 10.1017/jns.2017.66
  4. Timon, C,M.; Blain, R,J.; McNulty, B.; Kehoe, L.; Evans, K.; Walton, J. et al. The Development, Validation, and User Evaluation of Foodbook24: A Web-Based Dietary Assessment Tool Developed for the Irish Adult Population. J Med Internet Res. 2017, 19, doi: 10.2196/jmir.6407

Round 2

Reviewer 2 Report

I want to thank the authors for the changes made. No further comments.